# STRUCTURED STOCHASTIC GRADIENT MCMC

## ABSTRACT

Stochastic gradient Markov Chain Monte Carlo (SGMCMC) is considered the gold standard for Bayesian inference in large-scale models, such as Bayesian neural networks. Since practitioners face speed versus accuracy tradeoffs in these models, variational inference (VI) is often the preferable option. Unfortunately, VI makes strong assumptions on both the factorization and functional form of the posterior. In this work, we propose a new non-parametric variational approximation that makes no assumptions about the approximate posterior's functional form and allows practitioners to specify the exact dependencies the algorithm should respect or break. The approach relies on a new Langevin-type algorithm that operates on a modified energy function, where parts of the latent variables are averaged over samples from earlier iterations of the Markov chain. This way, statistical dependencies can be broken in a controlled way, allowing the chain to mix faster. This scheme can be further modified in a "dropout" manner, leading to even more scalability. By implementing the scheme on a ResNet-20 architecture, we obtain better predictive likelihoods and faster mixing time than full SGMCMC.

## 1 INTRODUCTION

There has been much recent interest in deep Bayesian neural networks (BNN) due to their reliable confidence estimates and generalization properties (Wilson & Izmailov, 2020; Jospin et al., 2020; Cardelli et al., 2019). BNNs rely on ensemble averages over model parameters typically obtained from Markov chain Monte Carlo (MCMC) algorithms, which contrasts to regular neural networks that depend on a single set of parameters. The sheer size of these models requires scalable MCMC approaches based on inexpensive stochastic gradients, of which stochastic gradient Markov chain Monte Carlo (SGMCMC) algorithms are the gold standard (Li et al., 2016; Welling & Teh, 2011; Patterson & Teh, 2013). These algorithms owe their scalability to approximating gradients via mini-batching.

The main downside of using SGMCMC algorithms is their slow mixing rates in high dimensions. An often faster alternative is variational inference (VI) algorithms that approximate the posterior with a simpler (typically factorized) distribution. This formulation results in an optimization problem that can be solved more efficiently using stochastic optimization (Blei et al., 2017; Zhang et al., 2018).

One downside of VI approximations is their ~~solid~~ distributional assumptions. A typical choice is to approximate the Bayesian posterior by a product of univariate Gaussian distributions. These distributional assumptions are frequently over-simplistic in high-dimensional models, where the posterior can be highly multi-modal and possibly heavy-tailed. Another downside is that the variational approximation typically underestimates the posterior variance, leading to poorly calibrated uncertainties and overfitting (Ormerod & Wand, 2010; Giordano et al., 2015; Zhang et al., 2018).

In this work, we derive a fundamentally new SGMCMC approach that takes inspiration from structured VI. While our approach remains a sampling algorithm resembling SGMCMC, we speed up the mixing time by systematically breaking posterior correlations. The resulting algorithm furthermore allows users to specify which posterior correlations to keep and which ones to break. It makes no assumptions on the functional form of the approximate posterior. We call our approach *structured SGMCMC* since it relies on a structured (i.e., only partially factorized) variational approximation of the posterior (Wainwright & Jordan, 2008).

In more detail, we derive the optimal variational distribution for a given posterior subject to factorization constraints by assuming a functional view on variational inference. We show how to sample from

this optimal distribution by running SGMCMC on a modified energy function. This energy function is obtained by marginalizing the model's joint distribution over previously generated samples from the Markov chain, leading to an approximate factorization over user-specified parameter groups. Further, we provide a more robust and computationally efficient approximation to the procedure that allows for interpolation between regular SGMCMC and our structured SGMCMC by taking inspiration from dropout techniques. Both methods are compatible with *any* Markovian SGMCMC algorithm, including Langevin dynamics and stochastic gradient Hamiltonian Monte Carlo.

In sum, our contributions are as follows:

- We propose a new approximate MCMC scheme running SGMCMC on a modified energy function, trading accuracy for speed. This setup effectively allows sampling from a fully joint posterior, a completely factorized posterior, and any in-between.

- We prove mathematically that the resulting scheme asymptotically generates samples from the best possible posterior approximation subject to user-specified factorization constraints between groups of parameters.

- We extend this scheme further by making it more scalable with a dropout-inspired approximation. This new scheme has a hyperparameter that enables a smooth interpolation between full SGMCMC and a "mean-field" version where all posterior correlations are broken.

- We show in both small and large scale experiments that our method well approximates posterior marginals and gives improved results over SGMCMC on Resnet-20 architectures on CIFAR-10, Fashion MNIST, and SVHN in terms of both runtime and final accuracy.

Our paper is structured as follows: Section 2 presents the related work to our proposal, Section 3 introduces preliminaries regarding the energy function and the stochastic gradient updates, Sections 4 and 5 derive our proposed methods, Section 6 details experiments and their results, and Section 7 contains our concluding thoughts.

## 2 RELATED WORK

Our work connects both to (stochastic) variational inference (Bishop, 2006; Hoffman et al., 2013; Ranganath et al., 2014; Blei et al., 2017; Zhang et al., 2018) and scalable MCMC (Welling & Teh, 2011; Chen et al., 2014; Ma et al., 2017; Zhang et al., 2020; Leimkuhler et al., 2019; Wenzel et al., 2020; Izmailov et al., 2021). For space limitations, we focus on the most related work at the intersection of both topics.

Among the earliest works to hybridize both approaches was (de Freitas et al., 2001) who constructed a variational proposal distribution in the Metropolos-Hastings step of MCMC. An improved approach to that was introduced in (Habib & Barber, 2018), where by introducing low-dimensional auxiliary variables they fit a more accurate approximating distribution. Other related advances to MCMC methods were proposed by Levy et al. (2017) who developed a method to train MCMC kernels with NNs, and Wang et al. (2018); Gong et al. (2018) who leveraged meta learning schemes in SGMCMC methods.

Most recent work focuses on connections between VI and stochastic gradient-based MCMC, or between VI and stochastic gradient descent (SGD). For example, Mandt et al. (2016; 2017) and Duvenaud et al. (2016) consider SGD as a type of variational inference, but their approaches did not attempt to close the gap to exact MCMC. Other works aim at explicitly interpolating between both methods. Domke (2017) proposes a divergence bound for hybridizing VI and MCMC, essentially by running Langevin dynamics on a tempered evidence lower bound (ELBO). Salimans et al. (2015) embody MCMC steps into the variational inference approximation. Ahn et al. (2012) improve stochastic gradient Langevin dynamics by leveraging the central limit theorem and using the estimated inverse Fisher information matrix to sample from the approximate posterior distribution. Rezende & Mohamed (2015) interpreted the path of an MCMC algorithm as a variational distribution, and then fitting parameters to tighten a variational bound. Recently, Hoffman & Ma (2020) interpreted (parametric) VI as approximate Langevin dynamics and showed that both algorithms have similar transient dynamics.

In contrast to all these approaches, our method is inspired by coordinate ascent variational inference (Bishop, 2006) but uses Langevin updates to generate samples from the target distribution that respects an imposed independence structure.

## 3  PRELIMINARIES

Variational inference (VI) approaches differ from MCMC in two regards: (1) they impose a structured (e.g., fully-factorized) approximation of the posterior for tractability, and (2) they often make parametric assumptions. Is it possible to construct a modified scheme that only relies on the assumption (1), inheriting the non-parametric nature of MCMC while breaking posterior correlations in a controlled manner? As follows, we will show how such a scheme can be realized. We will first derive a modified energy function for Langevin dynamics that we can sample from and then prove that its negative exponential results in the optimal posterior approximation subject to specified factorization constraints. Running SGMCMC algorithms on this energy function will consequently generate samples from this distribution.

Before we explain our new method, we introduce the setup and common notation. Given data $\mathcal{D} = \{(x_i, y_i)\}_{i=1,\dots,N}$, parameters $\theta$, a proper prior distribution $p(\theta)$, and a likelihood $p(\mathcal{D}|\theta) = \prod_{i=1}^{N} p(y_i|x_i, \theta)$, suppose we are interested in the corresponding posterior distribution $p(\theta|\mathcal{D}) \propto p(\mathcal{D}|\theta)p(\theta)$. A convenient representation of the posterior is as a Boltzmann distribution:

$$p(\theta|\mathcal{D}) \propto \exp\{-U(\theta)\} \text{ where } U(\theta) = -\log p(\theta, \mathcal{D}) = - \sum_{(x,y)\in\mathcal{D}} \log p(y|x, \theta) - \log p(\theta). \quad (1)$$

$U$ is typically referred to as the *posterior energy function*. Note that the posterior distribution is typically intractable due to the normalizing constant.

A popular approach for approximating the entire posterior distribution is by deploying Markov chain Monte Carlo (MCMC) algorithms. These methods work by producing an empirical distribution of samples in parameter space, often times through the use of a random walk. While being very accurate and having asymptotic guarantees, these methods are known to not scale well with respect to both data and parameters (Brooks et al., 2011; Geyer, 1992).

Stochastic gradient MCMC (SGMCMC) is a class of scalable MCMC algorithms that can produce posterior samples through gradients on minibatches of data. These algorithms are largely derived from discretized approximations of continuous-time diffusion processes. Examples of these algorithms include stochastic gradient Langevin dynamics (SGLD) (Welling & Teh, 2011), preconditioned SGLD (pSGLD) (Li et al., 2016), and stochastic gradient Hamiltonian Monte Carlo (SGHMC) (Chen et al., 2014).

As alluded to, the basis of SGMCMC algorithms is using a sampled minibatch of data $\tilde{\mathcal{D}}$ from $\mathcal{D}$ to produce an differentiable, unbiased estimate of the posterior energy function:

$$U(\theta) \approx \hat{U}(\theta; \tilde{\mathcal{D}}) = -\frac{N}{|\tilde{\mathcal{D}}|} \sum_{(x,y)\in\tilde{\mathcal{D}}} \log p(y|x, \theta) - \log p(\theta). \quad (2)$$

Once $\hat{U}$ is defined, it is fairly straight forward to generate new samples from the posterior distribution. For instance, the SGLD update is

$$\theta^{(t+1)} = \theta^{(t)} - \frac{\epsilon_t}{2}\nabla_\theta \hat{U}(\theta^{(t)}; \tilde{\mathcal{D}}_t) + \xi_t \quad \text{where} \quad \xi_t \sim \mathcal{N}(0, \epsilon_t I). \quad (3)$$

Similar rules for pSGLD and SGHMC can be found in the Supplement. All of these update rules produce a *chain* of samples up to time step $t$ that ultimately form an empirical distribution $\hat{p}^{(t)}(\theta|\mathcal{D})$. Should the algorithms converge, then $\lim_{t\to\infty} \hat{p}^{(t)}(\theta|\mathcal{D}) = p(\theta|\mathcal{D})$.

## 4  STRUCTURED SGMCMC

By design, SGMCMC methods produce a fully joint posterior distribution over parameters $\theta$. For models with a large number of parameters, this can lead to various complications due to the curse of

dimensionality. This is typically observed with slow convergence times and potentially unexplored parameter spaces. A viable solution is to break dependencies in the posterior distribution by leveraging ideas commonly used in variational inference (VI). This would reduce the number of various potential posterior correlations that the model would need to capture while sampling.

To achieve partial factorization, we must first partition $\theta$ into $M > 1$ distinct, **mutually independent** groups: $\theta_1, \ldots, \theta_M$. This partitioning structure is assumed to be known *a priori*. We will denote the distribution that respects this partitioning structure as $q(\theta) = \prod_{i=1}^{M} q_i(\theta_i)$. Similar to VI, we would like this distribution $q(\theta)$ to best approximate the true posterior distribution $p(\theta|\mathcal{D})$ according to some criteria, such as KL-divergence. This leads to a natural objective function to minimize:

$$J(q(\theta)) = D_{\text{KL}}(q(\theta)||p(\theta|\mathcal{D})) \equiv \mathbb{E}_{\theta \sim q}\left[\log \frac{q(\theta)}{p(\theta|\mathcal{D})}\right] \qquad (4)$$

The following Theorem 1 proves that there is a unique solution to the non-parametric KL minimization problem described in Eq. (4). To describe it, we compose $\theta = \{\theta_i, \tilde{\theta}_{\neg i}\}$ for any $i$ where $\tilde{\theta} \sim q$ and define a structured energy function:

$$U^{(S)}(\theta) = \sum_{i=1}^{M} U_i^{(S)}(\theta_i), \ \ \text{with } U_i^{(S)}(\theta_i) := \mathbb{E}_{\tilde{\theta} \sim q} U(\{\theta_i, \tilde{\theta}_{\neg i}\}) := -\mathbb{E}_{\tilde{\theta} \sim q} \log p(\theta_i, \tilde{\theta}_{\neg i}, \mathcal{D}). \quad (5)$$

That is, we first define the marginals $U_i^{(S)}(\theta_i)$, where we marginalize $U(\theta)$ with respect to all $q(\theta)$-factors except $q_i(\theta_i)$, and then sum up these marginals to define $U^{(S)}(\theta)$. A similar partial marginalization procedure is carried out for conjugate exponential family distributions in coordinate ascent VI (Bishop, 2006). Having a well-defined energy function $U^{(S)}$ allows us to use standard SGMCMC methods to approximate the posterior $q(\theta)$ with samples. This serves as the basis for our proposed algorithm that actually approximates this distribution $q(\theta)$, which will be discussed shortly.

**Theorem 1.** *The unique solution to the KL minimization problem given in Eq. 4 is given by the Boltzmann distribution* $q(\theta) \propto \exp\{-\sum_{i=1}^{M} U_i^{(S)}(\theta_i)\}$. Please refer to the Supplement for the proof.

In an ideal world, we would be able to use the findings of Theorem 1 directly in conjunction with algorithms like Langevin dynamics and Hamiltonian Monte Carlo to produce empirical distributions for $q$ using $U^{(S)}$ (Liu et al., 2019). However, this is intractable for two reasons: (1) these algorithms generally work only well with small amounts of data, and (2) more importantly, the marginals $U_i^{(S)}(\theta_i)$ do not have a closed-form solution but need to be approximated via samples from $q$. Luckily, since SGMCMC methods only need access to noisy estimates of $U^{(S)}$, we can run these algorithms on a stochastic estimate of Eq. (5),

$$U^{(S)}(\theta) \approx \hat{U}^{(S)}(\theta; \tilde{\mathcal{D}}) = \sum_{i=1}^{M} \mathbb{E}_{\tilde{\theta} \sim q} \hat{U}(\{\theta_i, \tilde{\theta}_{\neg i}\}; \tilde{\mathcal{D}}), \qquad (6)$$

where $\hat{U}(\cdot)$ is defined in Eq. (2). In practice, at timestep $t$ for $i = 1, \ldots, M$ we estimate $\mathbb{E}_{\tilde{\theta} \sim q} \hat{U}(\{\theta_i, \tilde{\theta}_{\neg i}\}; \tilde{\mathcal{D}}_t)$ with a Monte Carlo approximation. In place of $\tilde{\theta}$, we use a single sample of $\tilde{\theta}^{(t)}$ taken from the current approximate distribution $\hat{q}^{(t)}$ which is composed of samples from previous timesteps (i.e., a uniform distribution over $\{\theta^{(1)}, \theta^{(2)}, \ldots, \theta^{(t)}\}$). This leads to the following update step for structured SGLD (S-SGLD):

$$\theta^{(t+1)} = \theta^{(t)} - \frac{\epsilon_t}{2} \nabla_\theta \hat{U}^{(S)}(\theta; \tilde{\mathcal{D}}) + \xi_t \text{ where } \xi_t \sim \mathcal{N}(0, \epsilon_t I). \qquad (7)$$

Similar rules for structured variants of pSGLD (S-pSGLD) and SGHMC (S-SGHMC) can be found in the Supplement. Additionally, the full procedure for structured SGMCMC (S-SGMCMC) can be seen in Algorithm 2.

**Remark** Since $\nabla_\theta \hat{U}^{(S)}$ is an unbiased estimator for $U^{(S)}$, we are guaranteed to converge to $q$ from sampling with S-SGMCMC with sufficiently decreasing learning rates so long as we are in a *stationary state*. While it is unlikely to have the procedure initialize to a stationary state, we observe in practice that our scheme both tends to converge towards and remain in a stationary state. A general proof of convergence is outside the scope of this work and is left to follow-up research.

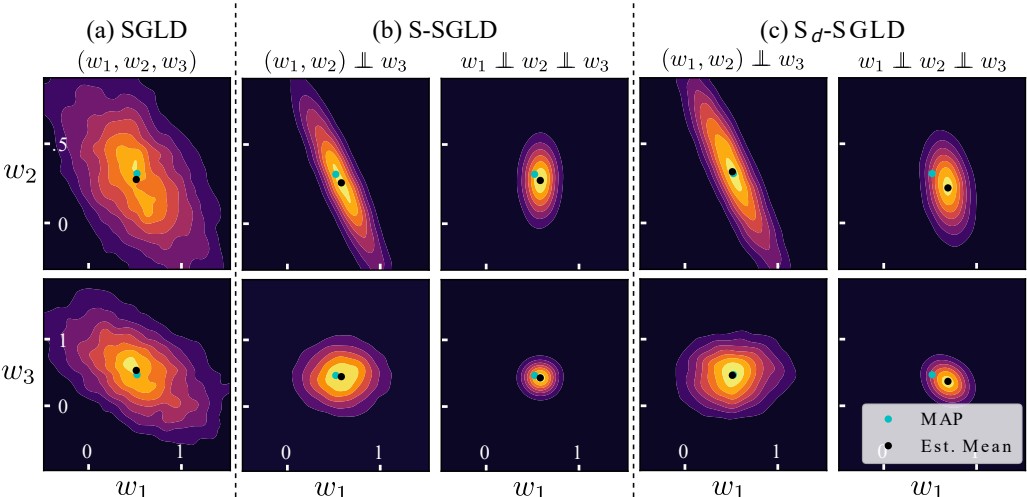

Figure 1: Sampled posterior distributions between $w_1$ & $w_2$ (top row) and $w_1$ & $w_3$ (bottom row) for a linear regression model. From left to right, (a) the first column has posterior distributions sampled with SGLD, (b) the second and third columns are sampled from S-SGLD, and (c) the fourth and fifth are sampled from $S_d$-SGLD. The imposed parameter structure for each are shown above the plots.

An example of S-SGMCMC can be seen in Fig. 1(a-b), which features the approximate posterior distributions of a linear regression model with three coefficients and with various independence structures imposed with S-SGLD: (a) joint dependence between $w_1, w_2$, and $w_3$; (b-left) dependence between $w_1$ and $w_2$ but independence between $w_3$ and the other coefficients; (b-right) fully factorized. Of note is that the bivariate posterior distributions appear to respect the imposed independence structure. Interestingly, it also appears that the variance shrinks as we induce these factorizations which is a commonly seen artifact when using VI.

## 5 STRUCTURED DROPOUT SGMCMC

While S-SGMCMC can successfully break dependencies between parameter groups, it does suffer computationally due to each parameter update scaling linearly with respect to $M$. This means that for a single new sample of $\theta$, the model's forward pass needs to be computed $M$ different times on the same batch of data $\tilde{\mathcal{D}}$, which can quickly become prohibitively expensive for deep models when $M$ is large. Ideally, we would prefer a method that both closely resembles the S-SGMCMC procedure and scales independently from the partitioning scheme. This section presents such a method that achieves this, which we call *structured dropout* SGMCMC ($S_d$-SGMCMC), as well as an informal motivation and derivation of the method. More formal details and a theorem proving both SGMCMC and S-SGMCMC are limiting cases for $S_d$-SGMCMC can be found in the Supplement.

The main motivation for this technique can be seen by recognizing that the composition $\{\theta_i^{(t)}, \tilde{\theta}_{\neg i}^{(t)}\}$ from Eq. (6) can be rewritten as a sum of masked values $r\theta^{(t)} + (1-r)\tilde{\theta}^{(t)}$ where $\tilde{\theta}^{(t)} \sim q^{(t)}$ and $r_j = \mathbb{1}(i = j)$ for $i = 1, \ldots, M$. We can decouple the computational scaling from the number of parameter groups $M$ by replacing the $M$ deterministic masks $r$'s with $K$ stochastically sampled masks $\tilde{r}$.[1] Doing so results in a slightly different energy function and minibatch loss to optimize:

$$\hat{U}^{(S_d)}(\theta^{(t)}; \tilde{\mathcal{D}}) \approx \frac{M}{K \mathbb{E}\left[\sum_{i=1}^{M} r_i\right]} \sum_{k=1}^{K} \hat{U}(\tilde{r}^{(t,k)}\theta^{(t)} + (1 - \tilde{r}^{(t,k)})\tilde{\theta}^{(t,k)}; \tilde{\mathcal{D}}) \tag{8}$$

where $\tilde{r}^{(t,k)}$ is the $k^{\text{th}}$ sample of $\tilde{r}$ for timestep $t$. A formal justification for Eq. (8) can be found in the Supplement. These energy function approximations lead to the following update step for structured

---

[1] $K$ is a hyperparameter that is chosen independent of $M$; however, both $M$ and the distribution of $\tilde{r}$ largely influence how small $K$ can be due to how they affect the variance of the gradient of the associated posterior energy function.

**Algorithm 2:** S-SGMCMC

**Input:** Initial sample $\theta^{(0)}$; parameter partitions $\theta_1, \ldots, \theta_M$; step sizes $\{\epsilon_t\}_{t=0,\ldots,T-1}$.

**Output:** $\hat{q}^{(T)}(\theta) := \{\theta^{(t)}\}_{t=1,\ldots,T}$

1 **for** $t = 0$ *to* $T - 1$ **do**
2      Sample minibatch $\tilde{\mathcal{D}}^{(t)} \subset \mathcal{D}$
3      **for** $i = 1$ *to* $M$ **do**
4          Sample $\tilde{\theta}_{\neg i}^{(t)} \sim \hat{q}_{\neg i}^{(t)}$
5          $\hat{U}_i^{(S,t)} = \hat{U}([\theta_i^{(t)}, \tilde{\theta}_{\neg i}^{(t)}]; \tilde{\mathcal{D}}^{(t)})$
6      **end**
7      $\nabla_\theta \hat{U}^{(S,t)} = \sum_{i=1}^{M} \nabla_\theta \hat{U}_i^{(S,t)}$
8      $\theta^{(t+1)} =$
         $\texttt{SGMCMC\_step}(\theta^{(t)}, \nabla_\theta \hat{U}^{(S,t)}, \epsilon_t)$
9 **end**
10 **return** $\hat{q}^{(T)}(\theta)$

Table 1: IAC and ESS metrics for CIFAR-10, SVHN, and FMNIST with various methods. Subscripts after method names refers to number of equally sized parameter groups, with $|\theta|$ meaning every parameter belongs to its own group. Best results are bolded.

| Method | CIFAR-10 | | SVHN | | FMNIST | |
|---|---|---|---|---|---|---|
| | IAC↓ | ESS↑ | IAC↓ | ESS↑ | IAC↓ | ESS↑ |
| pSGLD | 716 | **8.01** | 839 | 6.82 | 779 | 7.09 |
| S-pSGLD$_2$ | 600 | 7.44 | 840 | 6.80 | 740 | 7.55 |
| S-pSGLD$_4$ | 599 | 7.4 | 834 | 6.83 | 751 | 7.45 |
| S-pSGLD$_8$ | 709 | 6.41 | 857 | 6.67 | 776 | 7.24 |
| S$_d$-pSGLD$_{|\theta|}$ | **546** | **8.01** | **803** | **7.00** | **677** | **8.24** |
| SGHMC | 727 | 7.94 | 858 | 6.59 | 795 | 6.83 |
| S-SGHMC$_2$ | 583 | 7.49 | 949 | 5.74 | 928 | 5.67 |
| S-SGHMC$_4$ | 624 | 7.03 | 961 | 5.66 | 915 | 5.77 |
| S-SGHMC$_8$ | 904 | 4.97 | 1056 | 5.30 | 1142 | 4.87 |
| S$_d$-SGHMC$_{|\theta|}$ | 584 | 7.7 | 828 | 6.56 | 782 | 7.08 |

dropout variant of SGLD (S$_d$-SGLD):

$$\theta^{(t+1)} = \theta^{(t)} - \frac{\epsilon_t}{2} \nabla_\theta \hat{U}^{(S_d)}(\theta; \tilde{\mathcal{D}}) + \xi_t \text{ where } \xi_t \sim \mathcal{N}(0, \epsilon_t I). \tag{9}$$

The corresponding update rules for the structured dropout variants for pSGLD (S$_d$-pSGLD) and SGHMC (S$_d$-SGHMC) are defined in the Supplement. The exact procedure for generating samples of the approximate posterior $\hat{q}^{(t)}$ using structured dropout SGMCMC (S$_d$-SGMCMC) can also be found in the Supplement.

An example of this method (specifically S$_d$-SGLD with $\tilde{r}_i \overset{iid}{\sim}$ Bernoulli(0.5) and $K = 4$) used on a linear regression model can be seen in Fig. 1(c). Of note, we can see that the dropout variant largely respects the independence structure imposed, but maybe not as strictly as the exact S-SGLD method seen in Fig. 1(b). Additionally, the posterior variance also seems to have shrunk similarly to S-SGLD when compared against SGLD.

**Masking Distribution** Should $\tilde{r}_i \overset{iid}{\sim}$ Bernoulli($\rho$), alongside a structure that factorizes by activation components, then the method starts to resemble dropout with rate $\rho$ (Srivastava et al., 2014). The main difference being that instead of replacing a parameter value with 0 it is replaced with a sample from the approximate posterior distribution at time $t$: $\hat{q}^{(t)}$. While a Bernoulli distribution for $\tilde{r}$ is a natural choice, there are other distributions that can be chosen as well. For instance, $\tilde{r}_i \overset{iid}{\sim} \mathcal{N}(0, 1)$ or $\tilde{r}_i \overset{iid}{\sim}$ Beta($\alpha, \beta$) are both viable distributions and can be seen as analog to Gaussian and Beta-dropout respectively (Srivastava et al., 2014; Liu et al., 2019). Our experiments will largely focus on sampling $\tilde{r}$ from Bernoulli and uniform over $[0, 1]$ (equivalent to Beta(0.5, 0.5)) distributions.

## 6 EXPERIMENTS

**Overview** In this section we evaluate our proposed approach on various models and datasets. Section 6.1 investigates the impact of the variational approximation on the algorithms' mixing and autocorrelation times using a fully-connected network architecture on MNIST (LeCun et al., 2010). Section 6.2 studies our methods with ResNet-20 (He et al., 2016) on CIFAR-10 (Krizhevsky et al., 2009), SVHN (Netzer et al., 2011), and Fashion MNIST (Xiao et al., 2017) and compares them for their accuracy and mixing time. Our experiments reveal that the chains in our proposed methods mix faster than SGMCMC and achieve either comparable or even higher accuracies on average.

We have also conducted experiments on uncertainty visualization, where we tested the proposed methodology on predictive uncertainty estimation by deploying a two-layer fully connected network

on a toy dataset. The uncertainty experimental setup and results, along with more technical details for the other experiments, can be found in the Appendix.

**Metrics**   The primary predictive metric of interest use to evaluate our proposal is classification accuracy. We take the average of an ensemble of 100 models whose weights are sampled from the past samples of the parameters chains in order to calculate the accuracy. Additionally, we also monitor the mixing time of the chains of our methods with both integrated autocorrelation time (IAC) (Sokal, 1997; Goodman & Weare, 2010) and effective sample size (ESS) (Geyer, 1992). IAC measures the correlation between samples in a chain and, in turn, describe the inefficiency of a MCMC algorithm. IAC is computed as $\tau_f = \sum_{\tau=-\infty}^{\infty} \rho_f(\tau)$ where $\rho_f$ is the normalized autocorrelation function of the stochastic process that generated the chain for $f$ and is calculated as $\hat{\rho}_f(\tau) = \hat{c}_f(\tau)/\hat{c}_f(0)$; where $\hat{c}_f(\tau) = \frac{1}{N-\tau} \sum_{n=1}^{N-\tau} (f_n - \mu_f)(f_{n+\tau} - \mu_f)$ and $\mu_f = \frac{1}{N} \sum_{n=1}^{N} f_n$. We note that we calculated $\hat{c}_f(\tau)$ with a fast Fourier transform as it is more computationally efficient than using the direct sum. ESS measures how many independent samples would be equivalent to a chain of correlated samples and is calculated as $n_{\text{eff}} = \frac{n}{1+(n-1)p}$, where $n$ is the number of samples and $p$ is the autocorrelation.[2] We note that a model with higher ESS and lower IAC have faster mixing time. Please see the Appendix for the detailed implementation details and experimental setup for the metrics and our models.

## 6.1   DROPOUT RATE & GROUP SIZE INVESTIGATION

The aim of this set of experiments is to study the effects that the number of independent parameter groups (or alternatively, the amount of allowed posterior correlations) has on accuracy and mixing time when using our proposed methods. We compare pSGLD, S-pSGLD, and $S_d$-pSGLD a Bernoulli($\rho$) masking distribution with dropout rates $\rho \in \{0.1, 0.3, 0.5\}$ on a fully-connected neural network with 2 hidden layers, with 50 hidden units each, trained and evaluated with MNIST using the standard train and test split. The model has 42,200 parameters in total. For S-pSGLD and $S_d$-pSGLD, these parameters are evenly distributed into $M$ groups where $M$ ranges from 4 to 42,200. Accuracy, IAC, and ESS are reported in Fig. 2 using 100,000 posterior samples after a 150,000 burn in period. More details on the implementation of the model regarding training and evaluation can be found in the Appendix.

As shown in Fig. 2(a), for S-pSGLD we observe that as we increase the number of groups the accuracy drops dramatically whereas $S_d$-pSGLD's accuracy improves slightly and then remains fairly stable. In the best case, $S_d$-pSGLD achieves an accuracy of 96.3% with 32 groups and dropout rate of 0.5 which outperforms pSGLD with accuracy of 94.2%. We speculate that the dropout-like behavior is beneficial for regularizing the model (much like normal dropout), hence the improved accuracy across all dropout rates. Similarly, a single sample used for the Monte Carlo estimate in S-SGMCMC may not be enough as the number of groups $M$ increase; however, increasing the number of samples in this scenario is infeasible due to S-SGMCMC scaling as $\mathcal{O}(M)$.

Fig. 2(b-c) portrays the comparison between number of groups and mixing time metrics IAC and ESS. As the number of groups gradually increase, we note that S-pSGLD mixes faster, as does $S_d$-pSGLD to lesser and lesser degrees as $\rho$ increases. This behavior is to be expected due to Theorem 2, with $S_d$-pSGLD exhibiting mixing times more similar to pSGLD when $\rho = 0.5$ and more similar to S-pSGLD when $\rho = 0.1$.

## 6.2   SYSTEMATIC COMPARISON ON REAL-WORLD DATA

The goal of these experiments is to test the proposed methodology on larger-scale datasets which mimic real-world data: CIFAR-10, SVHN, and FMNIST. We evaluate our methods on performance accuracy and on mixing times of the chains. We employ ResNet-20 for SVHN and FMNIST without any data augmentation to assess our methods. For CIFAR10 we employ the same data augmentation process as proposed in Cubuk et al. (2019). We evaluate the precision of the methods on accuracy over time and the overall mixing time of them on IAC and ESS with 2 base algorithms: pSGLD and SGHMC. For efficiency purposes we limited our scope to models with either fully joint posteriors or fully factorized. As such, for the latter we employed $S_d$-SGMCMC methods

---

[2]We used the TensorFlow implementation for ESS which uses the direct sum for the autocorrelation.

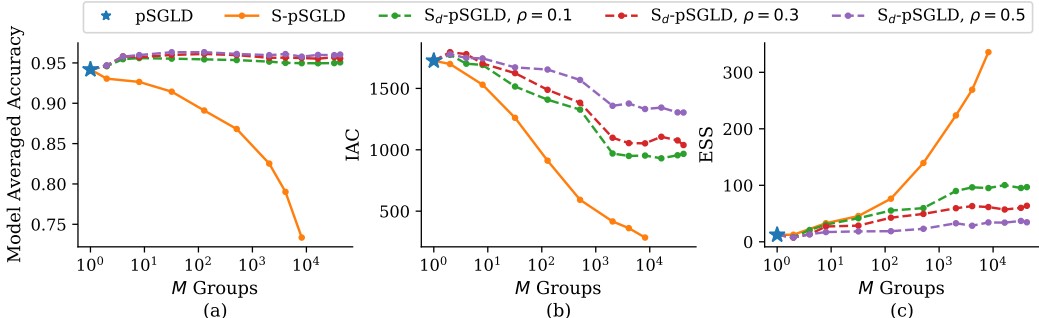

Figure 2: Comparison between pSGLD, S-pSGLD, and $S_d$-pSGLD of various numbers of parameter groups $M$ for (a) accuracy, (b) IAC, and (c) ESS. Right-most points on the plots represent models that have every parameter belonging to its own parameter group. S-pSGLD methods are were not able to be evaluated at such extreme values of $M$ due to computational scaling.

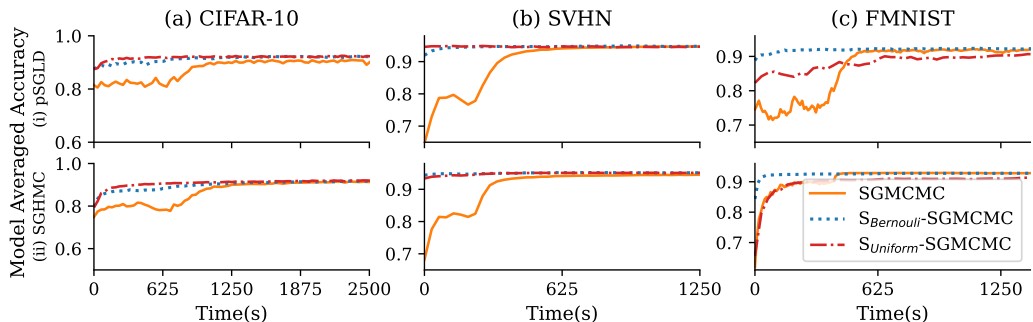

Figure 3: Comparison for (a) CIFAR-10, (b) SVHN and (c) FMNIST using ResNet-20 with (i) pSGLD and (ii) SGHMC sampling algorithms and their proposed variational variants for model averaged accuracy. Grid search was used to determine optimal hyperparameters for each method — more details can be found in the Appendix.

as S-SGMCMC would not be feasible with the amount of parameter groups present. Bernoulli($\rho$) and uniform masking distributions were investigated and are denoted as $S_{\text{Bernoulli}}$-SGMCMC and $S_{\text{Uniform}}$-SGMCMC respectively, with $\rho$ varying between datasets as determined by a hyperparameter search (detailed in the Appendix).

In Fig. 3 we observe how quickly the proposed methods and the baseline SGMCMC methods approach their optimum accuracy over the course of training. As is shown, $S_{\text{Bernoulli}}$-SGMCMC and $S_{\text{Uniform}}$-SGMCMC appear to achieve optimal accuracy values much faster than SGMCMC on all datasets and with all base sampling schemes. In some cases, the variational methods achieve better accuracy values than the baseline methods, as seen for CIFAR10 in Fig. 3.

**Mixing Time Comparisons** We further validated our findings from Section 6.1 by evaluating the IAC and ESS on larger datasets using various methods. Both pSGLD and SGHMC were used as base methods in conjunction with both S-SGMCMC and $S_d$-SGMCMC using a Bernoulli masking distribution. IAC and ESS were calculated for these methods using the latest 5,000 samples after sampling for 300 epochs; the results of which can be found in Table 1. For CIFAR-10, we see that $S_d$-SGMCMC with every parameter in a different group mixes the fastest against all other methods. Likewise, for SVHN and FMNIST, $S_d$-pSGLD with every parameter belonging to its own group mixes faster than all other methods. At times it does appear that increasing the number of parameter groups causes slower mixing time for S-SGMCMC. This could potentially be attributed to large variance in the gradients from using only a single sample per Monte Carlo estimate.

Table 2: MSE for various partitioning schemes on the parameters for a 2-layer fully-connected BNN.

| Partition Scheme | Wine | Housing | Obesity | Bike | Concrete | Airfoil |
|---|---|---|---|---|---|---|
| Random ($M = 3$) | 0.0454 ±0.001 | 0.0233 ±0.003 | 0.0232 ±0.005 | 0.0242 ±0.001 | 0.0226 ±0.003 | 0.0454 ±0.001 |
| By Layer | 0.0494 ±0.001 | 0.0236 ±0.003 | 0.0274 ±0.002 | 0.0247 ±0.001 | 0.0243 ±0.003 | 0.0494 ±0.001 |
| By Neurons | 0.0496 ±0.001 | 0.0233 ±0.003 | 0.0262 ±0.003 | 0.0247 ±0.001 | 0.0238 ±0.003 | 0.0496 ±0.001 |
| Fully-Factorized | 0.0478 ±0.002 | 0.0236 ±0.002 | 0.0227 ±0.002 | 0.025 ±0.001 | 0.0238 ±0.001 | 0.0478 ±0.002 |

## 6.3 EXPLORING PARTITIONING SCHEMES

This part of the study aims to explore the capabilities of the proposed methodology further. Here we explore different parameter partitioning schemes on regression datasets.

Here we present the results with different partitions on various regression datasets. We used 7 different datasets: the wine quality dataset (Cortez et al., 2009), the Boston housing dataset (Harrison Jr & Rubinfeld, 1978), the obesity levels dataset (Palechor & de la Hoz Manotas, 2019), the Seoul bike-sharing dataset (E et al., 2020; E & Cho, 2020), the concrete compressive strength dataset (Yeh, 1998), and the airfoil self-noise dataset (Brooks et al., 1989). For the evaluation we chose a simple fully connected network with two layers with 50 neurons each, and we use SGLD as an optimizer. As a performance metric we chose mean squared error (MSE). We did hyperparameter tuning with different learning rates and the final results are the means with the standard deviations of 5 runs with different seeds. We do not observe any specific systematic trends on the partitions, apart from the fact that in some cases random performs better. In that way the use of either random partitioning or the fully-factorized partitioning, where every parameter is in a different group appears to be a valid choice *a priori*; especially the latter since we have noted earlier the faster mixing times associated with this partitioning scheme. More details about the partitioning schemes experiments can be found in the Appendix.

## 7 CONCLUSIONS

In an attempt to hybridize MCMC and VI, we proposed S-SGMCMC: an approach that produces samples from an structured posterior by running SGMCMC on a modified energy function. The resulting Markov chain becomes asymptotically decoupled across user-specified groups of parameters, resulting in faster convergence. For better computational efficiency, we proposed $S_d$-SGMCMC: a further generalization of S-SGMCMC inspired by dropout. This extension allows interpolating between a SGMCMC algorithm and its corresponding S-SGMCMC method.

Our experimental results demonstrate that the proposed methods impose structure over posterior distributions, increase mixing times of the chains, and result in similar or better posterior predictive accuracies compared to SGMCMC on a variety of (deep) models. Our experimental evaluations have provided strong empirical evidence for the efficacy of our approach. We also showed that the proposed approach is compatible with various deep learning architectures, including ResNet-20, and various datasets.

Despite its proven capabilities, our proposed methodology does come with some limitations. Namely, for quick access our methods require keeping chains of samples on the GPU whereas the baseline SGMCMC methods can simply save samples to disk. Additionally, S-SGMCMC scales poorly with respect to the number of parameter groups. $S_d$-SGMCMC manages to break this dependency; however, it still requires slightly more compute than SGMCMC per sample, but it is comparable in wall clock time. Possible future work could focus on more theoretical analyses of S-SGMCMC, such as formal proofs of convergence.

## 8 ETHICS STATEMENT

The main focus of our work is to train models faster by decreasing the convergence time of their training phase. In this scope we are not aware of any ethical concerns of our research.

## 9 REPRODUCIBILITY

For this work we have made sure to guarantee reproducibility of the results. We provide all the technical details of our experiments and their implementations, like the hyperparameters, the data, the frameworks and the experimental setups. We have used only open source datasets that are easily accessible to the public. Finally we commit to release the code that we implemented for this work via a public repository.

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

## A  THEOREM 1

*Proof.* We begin with some preliminaries from the main text. Given data $\mathcal{D} = \{(x_i, y_i)\}_{i=1,\dots,N}$, parameters $\theta$, a proper prior distribution $p(\theta)$, and a likelihood $p(\mathcal{D}|\theta) = \prod_{i=1}^{N} p(y_i|x_i, \theta)$, suppose we are interested in the corresponding posterior distribution $p(\theta|\mathcal{D}) \propto p(\mathcal{D}|\theta)p(\theta)$. A convenient representation of the posterior is as a Boltzmann distribution:

$$p(\theta|\mathcal{D}) \propto \exp\{-U(\theta)\} \quad \text{where} \quad U(\theta) = - \sum_{(x,y) \in \mathcal{D}} \log p(y|x, \theta) - \log p(\theta). \tag{10}$$

$U$ is typically referred to as the *posterior energy function*. Note that the posterior distribution is typically intractable due to the normalizing constant.

We also write the equation for KL divergence from the main text:

$$J(q(\theta)) = D_{\text{KL}}(q(\theta)||p(\theta|\mathcal{D})) \tag{11}$$

$$\equiv \mathbb{E}_{\theta \sim q} \left[ \log \frac{q(\theta)}{p(\theta|\mathcal{D})} \right] \tag{12}$$

We then rewrite Eq. 4 as follows:

$$J(q(\theta)) = \mathbb{E}_{\theta \sim q} \left[ \log q(\theta) \right] - \mathbb{E}_{\theta \sim q} \left[ \log p(\theta, \mathcal{D}) \right] + C \tag{13}$$

$$= \mathbb{E}_{\theta_i \sim q_i} \left[ \log q_i(\theta_i) \right] + \sum_{i \neq j} \mathbb{E}_{\theta_j \sim q_j} \left[ \log q_j(\theta_j) \right] - \int \log p(\theta, \mathcal{D}) q_i(\theta_i) d\theta_i \prod_{i \neq j} q_j(\theta_j) d\theta_j + C \tag{14}$$

for some $i \in \{1, \dots, M\}$ where $\neg i := \{1, \dots, M\} \setminus \{i\}$ and $C = \log p(\mathcal{D})$. In order to find the optimal distribution that respects the factorization constraints imposed between parameter groups, we need to minimize this functional over $q$ — or rather every $q_i$. This is done by taking the functional derivative of $J$ with respect to $q_i$, setting it equal to zero, and solving for $q_i$:

$$\frac{\delta J(q(\theta))}{\delta q_i(\theta_i)} = \int \log p(\theta, \mathcal{D}) \prod_{i \neq j} q_j(\theta_j) d\theta_j - 1 - \log q_i(\theta_i) := 0 \tag{15}$$

$$\implies \log q_i(\theta_i) = \mathbb{E}_{\tilde{\theta}_{\neg i} \sim q_{\neg i}} \left[ \log p(\theta_i, \tilde{\theta}_{\neg i}, \mathcal{D}) \right] - 1 \tag{16}$$

$$\implies q_i(\theta_i) \propto \exp \left\{ \mathbb{E}_{\tilde{\theta}_{\neg i} \sim q_{\neg i}} \left[ \log p(\theta_i, \tilde{\theta}_{\neg i}, \mathcal{D}) \right] \right\}. \tag{17}$$

By defining the energy $U_i^{(S)}(\theta_i) = -\mathbb{E}_{\tilde{\theta}_{\neg i} \sim q_{\neg i}} \left[ \log p(\theta_i, \tilde{\theta}_{\neg i}, \mathcal{D}) \right]$, we realize that by minimizing the KL-divergence in Eq. 4, the approximate posterior distribution $q = \prod_{i=1}^{M} q_i$ takes the form of a Boltzmann distribution as in Eq. 1 with $U^{(S)}(\theta) = \sum_{i=1}^{M} U_i^{(S)}(\theta_i)$.

It remains to be shown that the solution is unique. To this end, we refer to the convexity of the KL divergence in function space (Cover & Thomas, 2001). This implies that the stationary point of the KL is indeed a global optimum and unique. □

## B  DERIVING $U^{(S_d)}$

With just a slight shift in perspective, it is actually possible to further generalize $U^{(S)}$ (and consequently S-SGMCMC) to produce a broader class of approximate sampling algorithms. This is done

by first noting that $U^{(S)}$ can be represented with a scaled double-expectation:

$$U^{(S)}(\theta) = -\frac{M}{\mathbb{E}_{r \sim p^{(S)}}\left[\sum_{i=1}^{M} r_i\right]} E_{r \sim p^{(S)}} E_{\tilde{\theta} \sim q}\left[\log p(r\theta + (1-r)\tilde{\theta}, \mathcal{D})\right] \tag{18}$$

where $p^{(S)}(r) = \text{Cat}(r; M^{-1}, \ldots, M^{-1})$ and $(r\theta + (1-r)\tilde{\theta})_i$ is equal to $\theta_i$ if $r_i = 1$ and $\tilde{\theta}_i$ otherwise for $i = 1, \ldots, M$. Note that this is constructed in this manner specifically so that $U^{(S)}$ remains differentiable with respect to $\theta$. Also note that though the denominator appears superfluous as $\mathbb{E}_{r \sim p^{(S)}}[\sum_{i=1}^{M} r_i] = 1$, it is necessary for certain theoretic properties, as seen in Theorem 2.

By replacing $p^{(S)}$ with a more flexible distribution, we can further generalize and encapsulate different energy functions to sample from. One such choice is $p^{(S_d)}(r; \rho) :\propto \prod_{i=1}^{M} \text{Bern}(r_i; \rho) \mathbb{1}(\sum_{i=1}^{M} r_i > 0)$ with $\rho \in (0, 1)$.[3] Substituting $p^{(S)}$ for $p^{(S_d)}$ in Eq. (18) yields a new energy function that we will refer to as $U^{(S_d)}$. We note that this choice in distribution leads to a dropout-like behavior (Nalisnick et al., 2019; Srivastava et al., 2014), where the composition of model parameters as $r\theta + (1-r)\tilde{\theta}$ leads to each parameter group $\theta_i$ having a probability of approximately $\rho$ to be used in a prediction and a $(1 - \rho)$ probability of being replaced by $\tilde{\theta}_i$ from the approximate posterior (in traditional dropout, $\theta_i$ would instead be replaced with 0). Likewise, we will denote methods that use this energy function for sampling as structured dropout SGMCMC ($S_d$-SGMCMC) with different variants all sharing the same $S_d$ prefix (e.g. $S_d$-SGHMC).

In practice, the double-expectation in $U^{(S_d)}$ is jointly approximated using a Monte Carlo estimate with $K$ samples. This leads to Eq. (8) in the main paper. We note that by approximating $U^{(S_d)}$ in this way, computing a gradient no longer scales on the order of $\mathcal{O}(M)$, but rather $\mathcal{O}(K)$. This means that the choice of structure imposed on the posterior distribution remains independent of computing resources. As such, configurations with large amounts of parameter groups are typically only feasible when using $S_d$-SGMCMC as S-SGMCMC would use too much memory and/or compute per sample.

## C  THEOREM 2

**Theorem 2.** *For a given set of parameters $\theta$ partitioned into $M$ groups, under minor assumptions (i) $U^{(S_d)} \to U$ as $\rho \to 1$ and (ii) $U^{(S_d)} \to U^{(S)}$ as $\rho \to 0$. Thus, distributions approximated by $S_d$-SGMCMC lie on a continuum with those generated by S-SGMCMC at one extreme and with those from SGMCMC at the other.*

*Proof.* Assume an arbitrary $\theta$, $\mathcal{D}$, $n \in \mathbb{N}$, and that $\mathbb{E}_{\tilde{\theta} \sim q}\left[\log p(r\theta + (1-r)\tilde{\theta}, \mathcal{D})\right]$ exists for $r \in \mathcal{R}$. As an aside, this proof assumes that $p^{(S_d)}(r; \rho) :\propto \prod_{i=1}^{M} \text{Bern}(r_i; \rho) \mathbb{1}(\sum_{i=1}^{M} r_i > 0)$ with $\rho \in (0, 1)$; however, the theorem still holds an arbitrary $p^{(S_d)}$ so long as the mean approaches 1 and variance approaches 0 as $n \to \infty$.

(i) Let $r^{(n)} \sim p^{(S_d)}(\rho_n)$ where $\forall_n \rho_n \in (0, 1)$ and $\rho_n \to 1$. It follows that $r^{(n)} \to \{1\}^M$ as $n \to \infty$ in distribution (see Lemma 1 in Supplement). Due to bounded and finite support $\mathcal{R}$, we find the following:

$$U^{(S_d)}(\theta) = -\frac{M}{\mathbb{E}_{r \sim p^{(S_d)}}[\sum_{i=1}^{M} r_i]} \sum_{r \in \mathcal{R}} p^{(S_d)}(r; \rho_n) \mathbb{E}_{\tilde{\theta} \sim q}\left[\log p(r\theta + (1-r)\tilde{\theta}, \mathcal{D})\right] \tag{19}$$

$$\to -\frac{M}{M} \sum_{r \in \mathcal{R}} \mathbb{1}(\forall_i r_i = 1) \mathbb{E}_{\tilde{\theta} \sim q}\left[\log p(\theta, \mathcal{D})\right] \text{ as } n \to \infty \tag{20}$$

$$= -\log p(\theta, \mathcal{D}) = U(\theta) \tag{21}$$

(ii) Let $r^{(n)} \sim p^{(S_d)}(\rho_n)$ where $\forall_n \rho_n \in (0, 1)$ and $\rho_n \to 0$. It follows that $r^{(n)} \to r \sim \text{Cat}(M^{-1}, \ldots, M^{-1})$ as $n \to \infty$ in distribution (see Lemma 2 in Supplement). Due to bounded and

---

[3]Other choices of distribution that are well justified include any with support over $[0, 1]^M$ and with measure 0 over $\{0\}^M$. Exploring the effects these distributions have are an interesting line of future inquiry.

finite support $\mathcal{R}$, we find the following:

$$U^{(S_d)}(\theta) = -\frac{M}{\mathbb{E}_{r \sim p^{(S_d)}}[\sum_{i=1}^{M} r_i]} \sum_{r \in \mathcal{R}} p^{(S_d)}(r; \rho_n) \mathbb{E}_{\tilde{\theta} \sim q} \left[ \log p(r\theta + (1-r)\tilde{\theta}, \mathcal{D}) \right] \tag{22}$$

$$\rightarrow -\frac{M}{1} \sum_{r \in \mathcal{R}} \frac{\mathbb{1}(\sum_{i=1}^{M} r_i = 1)}{M} \mathbb{E}_{\tilde{\theta} \sim q} \left[ \log p(r\theta + (1-r)\tilde{\theta}, \mathcal{D}) \right] \text{ as } n \rightarrow \infty \tag{23}$$

$$= -\sum_{i=1}^{M} \mathbb{E}_{\tilde{\theta} \sim q}[\log p([\theta_i, \tilde{\theta}_{\neg i}, \mathcal{D})] = U^{(S)}(\theta) \tag{24}$$

$$\square$$

For both Lemmas 1 and 2, let

$$p^{(S_d)}(r; \rho) = \frac{\rho^{\sum_{i=1}^{M} r_i}(1-\rho)^{M - \sum_{i=1}^{M} r_i}}{1 - (1-\rho)^M} \mathbb{1}(\forall_i r_i \in \{0, 1\}) \mathbb{1} \left( \sum_{i=1}^{M} r_i > 0 \right) \tag{25}$$

**Lemma 1.** *For $r^{(n)} \sim p^{(S_d)}(\rho_n)$, $\rho_n \in (0, 1)$ and $n \in \mathbb{N}$, if $\rho_n \rightarrow 1$ as $n \rightarrow \infty$ then $r^{(n)} \rightarrow r \sim \delta(\{1\}^M)$ in distribution as $n \rightarrow \infty$.*

*Proof.*

$$p^{(S_d)}(r = \{1\}^M; \rho_n) = \frac{\rho_n^M(1 - \rho_n)^0}{1 - (1 - \rho_n)^M} \tag{26}$$

$$\rightarrow 1 \text{ as } n \rightarrow \infty \tag{27}$$

$$\implies r^{(n)} \rightarrow \delta(\{1\}^M) \text{ in distribution.} \tag{28}$$

$$\square$$

**Lemma 2.** *For $r^{(n)} \sim p^{(S_d)}(\rho_n)$, $\rho_n \in (0, 1)$ and $n \in \mathbb{N}$, if $\rho_n \rightarrow 0$ as $n \rightarrow \infty$ then $r^{(n)} \rightarrow r \sim Cat(M^{-1}, \ldots, M^{-1})$ in distribution as $n \rightarrow \infty$.*

*Proof.* Let $i \in \{1, \ldots, M\}$.

$$p^{(S_d)}(r_i = 1, r_{\neg i} = 0; \rho_n) = \frac{\rho_n(1 - \rho_n)^{M-1}}{1 - (1 - \rho_n)^M} \tag{29}$$

$$\text{l'Hôpital's Rule} \quad \overset{H}{=} \frac{(1 - \rho_n)^{M-1} + \rho_n(M-1)(1 - \rho_n)^{M-2}}{M(1 - \rho_n)^{M-1}} \tag{30}$$

$$\rightarrow \frac{1}{M} \text{ as } n \rightarrow \infty \tag{31}$$

Since the resulting probabilities sum to 1, this implies that $r^{(n)} \rightarrow r \sim Cat(M^{-1}, \ldots, M^{-1})$ in distribution as $n \rightarrow \infty$. $\square$

## D DERIVING $U^{(S_d)}$

To derive $U^{(S_d)}$, we must first start with a shift in perspective on how $U^{(S)}$ is represented. We will rewrite the function in the following way:

$$U^{(S)}(\theta) = -\sum_{i=1}^{M} \mathbb{E}_{\theta_{\neg i} \sim q_{\neg i}}[\log p([\theta_i, \theta_{\neg i}], \mathcal{D})] \tag{32}$$

$$= -\frac{M}{\mathbb{E}_{r \sim p^{(S)}}[\sum_{i=1}^{M} r_i]} \mathbb{E}_{r \sim p^{(S)}} \mathbb{E}_{\tilde{\theta} \sim q} \left[ \log p(r\theta + (1-r)\tilde{\theta}, \mathcal{D}) \right] \tag{33}$$

where $p^{(S)}$ is a $M$-dimensional categorical distribution with uniform weights $M^{-1}$ and $p(r\theta + (1 - r)\tilde{\theta}, \mathcal{D})$ is the joint probability of parameters taking values of $r\theta + (1 - r)\tilde{\theta}$ and data $\mathcal{D}$.[4]

We note that changing the distribution of $r$ leads to different energy functions to sample from. One such choice is to have $p^{(S_d)}(r; \rho) \propto \rho^{\sum_{i=1}^{M} r_i}(1 - \rho)^{M - \sum_{i=1}^{M} r_i}\mathbb{1}(\forall_i r_i \in \{0, 1\})\mathbb{1}(\sum_{i=1}^{M} r_i > 0)$ for $\rho \in (0, 1)$. Note that this is identical to $r_i \overset{iid}{\sim} \text{Bernoulli}(\rho)$ conditional to $\sum_{i=1}^{M} r_i > 0$. Let the support of $p^{(S_d)}$ be denoted as $\mathcal{R} = \{0, 1\}^M \setminus \{0\}^M$. This leads to the following energy function:

$$U^{(S_d)}(\theta) = -\frac{M}{\mathbb{E}_{r \sim p^{(S_d)}}[\sum_{i=1}^{M} r_i]}\mathbb{E}_{r \sim p^{(S_d)}}\mathbb{E}_{\tilde{\theta} \sim q}\left[\log p(r\theta + (1 - r)\tilde{\theta}, \mathcal{D})\right]. \tag{34}$$

In practice, a few approximations are made to compute the corresponding $U^{(S_d)}$. Firstly, we approximate $p^{(S_d)}$ with an $M$-dimensional Bernoulli($\rho$) distribution as the difference is minute when $M\rho$ is large. Secondly, the outer expectation in Eq. (34) is approximated with a Monte Carlo estimate of $K$ samples. The inner expectation is also approximated with a Monte Carlo estimate using the latest approximate posterior $\hat{q}^{(t)}$. However, just like for S-SGMCMC, only a single sample is used. This further leads to:

$$U^{(S_d)}(\theta^{(t)}; \tilde{\mathcal{D}}) = -\frac{1}{K\rho}\sum_{k=1}^{K} U(r^{(t,k)}\theta^{(t)} + (1 - r^{(t,k)})\tilde{\theta}^{(t,k)}; \tilde{\mathcal{D}}) \tag{35}$$

# E    ALGORITHM FOR $S_d$-SGMCMC

The procedure for $S_d$-SGMCMC can be seen in Algorithm 3.

---

**Algorithm 3:** $S_d$-SGMCMC

**Input:** Initial sample $\theta^{(0)}$; parameter partitions $\theta_1, \ldots, \theta_M$; data set $\mathcal{D}$; initial auxiliary statistics $\xi^{(0)}$; step sizes $\{\epsilon_t\}_{t=1,\ldots,T}$; masking distribution $p^{(S_d)}$; dropout iterations $K$.
**Output:** $\hat{q}^{(T)}(\theta) := \{\theta^{(t)}\}_{t=1,\ldots,T}$

1  **for** $t = 0$ *to* $T - 1$ **do**
2      Sample minibatch $\tilde{\mathcal{D}}^{(t)} \subset \mathcal{D}$
3      **for** $k = 1$ *to* $K$ **do**
4          Sample masks $r_1^{(t,k)}, \ldots, r_M^{(t,k)} \sim p^{(S_d)}$
5          Sample $\tilde{\theta}^{(t,k)} \sim \hat{q}^{(t)}$
6          $\theta^{(t,k)} = [r_i^{(t,k)}\theta_i^{(t)} + (1 - r_i^{(t,k)})\tilde{\theta}_i^{(t,k)}]_{i=1,\ldots,M}$
7          $\hat{U}_k^{(S_d,t)} = \hat{U}(\theta^{(t,k)}; \tilde{\mathcal{D}}^{(t)})$
8      **end**
9      $\nabla_\theta \hat{U}^{(S_d,t)} = \frac{M}{K\mathbb{E}_{r \sim p^{(S_d)}}[\sum_{i=1}^{M} r_i]}\sum_{k=1}^{K}\nabla_\theta\hat{U}_k^{(S_d,t)}$
10      $\theta^{(t+1)}, \xi^{(t+1)} = \texttt{SGMCMC\_step}(\theta^{(t)}, \nabla_\theta\hat{U}^{(S_d,t)}, \xi^{(t)}, \epsilon_t)$
11  **end**
12  **return** $\hat{q}^{(T)}(\theta)$

---

[4]$r\theta + (1 - r)\tilde{\theta}$ is a slight abuse of notation that is meant to represent masking out $\theta_i$ when $r_i = 0$ and masking out $\tilde{\theta}_i$ when $r_i = 1$.

## F  SGMCMC UPDATE RULES

The update rules for SGLD, pSGLD, and SGHMC are defined as follows:

$$\text{SGLD}\ \ \theta^{(t+1)} = \theta^{(t)} - \frac{\epsilon_t}{2}\nabla_\theta \hat{U}(\theta^{(t)}) + \mathcal{N}(0, \epsilon_t I) \tag{36}$$

$$\text{pSGLD}\ \ \theta^{(t+1)} = \theta^{(t)} - \frac{\epsilon_t}{2}\left[ R(\theta^{(t)})\nabla_\theta \hat{U}(\theta^{(t)}) + \sum_\theta \nabla_\theta R(\theta^{(t)}) \right] + \mathcal{N}(0, \epsilon_t R(\theta^{(t)})) \tag{37}$$

$$\text{SGHMC}\ \ \theta^{(t+1)} = \theta^{(t)} + \epsilon_t M^{-1} m^{(t+1)} \tag{38}$$

$$m^{(t+1)} = (1 - \gamma\epsilon_t M^{-1})m^{(t)} - \epsilon_t \nabla_\theta \hat{U}(\theta^{(t)}) + \mathcal{N}(0, 2\gamma - \epsilon_t \hat{V}(\theta^{(t)})) \tag{39}$$

where $\epsilon_t$ is the step size at time step $t$, $R(\cdot)$ and $M$ are preconditioners, $\gamma \geq 0$ is a friction term, and $\hat{V}(\cdot)$ is an estimate of the covariance induced by the stochastic gradient.[5]

The update rules for the S-SGMCMC variants are similarly defined as Eqs. 36-39 but all instances of $\hat{U}(\theta^{(t)})$ are replaced with $\hat{U}^{(S)}(\theta^{(t)})$. Likewise, replacing with $\hat{U}^{(S_d)}(\theta^{(t)})$ yields the $S_d$-SGMCMC variants.

## G  ABLATION STUDY

This subsection aims to further explore the capabilities of the proposed methodology. More specifically, we ~~visualize uncertainty for a two-layer Fully Connected Network and~~ experiment with various parameter partitions.

**Parameter Partitions.**  We tested our proposal with four partitioning schemes on a 2 layer with 50 neurons fully connected network on a regression task. The partitioning schemes that we used are the following: (a) the parameters are split into 3 groups randomly, (b) the parameters are split by layer(3 layers, 1 input and 2 hidden), (c) by activating neurons inside the layers and (d) every parameter belongs in each own group. We used 7 different datasets: the wine quality datsetCortez et al. (2009), the Boston housing datasetHarrison Jr & Rubinfeld (1978), the obesity levels datasetPalechor & de la Hoz Manotas (2019), the Seoul bike-sharing datasetE et al. (2020); E & Cho (2020), the concrete compressive strength datasetYeh (1998), and the airfoil self-noise datasetBrooks et al. (1989). Every dataset was split into 75% training data, 10% validation data, and 15% test data. We trained the model on training set and validated it in the validation set with an early stoppage. For every dataset and every partitioning scheme we used the learning rates: 1e-3,1e-4,1e-5,1e-6,1e-7 for hyperparameter tuning. For each combination of partition and dataset, we chose the learning rate that provides the best accuracy score on the test set. In this case, as an accuracy score, we used the Mean Squared Error. The final learning rates that we used are presented in Table 3.

| Partition Scheme | Wine | Housing | Obesity | Bike | Concrete | Airfoil |
|---|---|---|---|---|---|---|
| Random ($M = 3$) | 1e-3 | 1e-5 | 1e-5 | 1e-4 | 1e-5 | 1e-4 |
| By Layer | 1e-3 | 1e-5 | 1e-4 | 1e-4 | 1e-5 | 1e-4 |
| By Neurons | 1e-3 | 1e-5 | 1e-4 | 1e-4 | 1e-5 | 1e-5 |
| Fully-Factorized | 1e-5 | 1e-4 | 1e-5 | 1e-4 | 1e-4 | 1e-3 |

Table 3: Best Learning Rates for various partitioning schemes on multiple regression datasets.

---

[5]Note that we abuse notation in Eqs. 36-39 where the addition of $\mathcal{N}(\mu, \Sigma)$ denotes the addition of a normally distributed random variable with mean $\mu$ and covariance $\Sigma$.

# H    DETAILS ON EXPERIMENTS

## H.1    QUALITATIVE REGRESSION EXPERIMENTS

First, we aim to showcase qualitative differences in the empirical posterior distributions generated by a baseline SGMCMC algorithm and our proposed variants. To do so, we consider a regression task where 100 randomly sampled three-dimensional covariates $\{\vec{x}_i = [x_{i,1}, x_{i,2}, x_{i,3}]^T\}_{i=1,\ldots,100}$ are used to sample response values $y_i \sim \mathcal{N}(\vec{w}^T \vec{x}_i + b, \sigma^2)$ where $\vec{w} = [w_1, w_2, w_3]^T = [1.5, -0.8, 1.3]^T$, $b = 0.5$, and $\sigma^2 = 1$. More details on the generation process for $\vec{x}$ can be found in the Supplement.

We choose to fit a linear regression model of the same form as the generation process. $\sigma^2$ is assumed to be known. Thus, $\theta = [w_1, w_2, w_3, b]$. A standard normal distribution is used as the prior for each parameter. Due to conjugacy, the posterior distribution can be calculated analytically. As such, the MAP is roughly $\hat{\theta}_{\text{MAP}} \approx [0.52, 0.31, 0.47, 0.84]$.

The approximated posterior distributions for $\theta$ are found using SGLD, S-SGLD, and $\text{S}_d$-SGLD. For the latter two sampling schemes, two parameter partitions are tested: (i) two groups of parameters where $\theta_1 = [w_1, w_2]$ and $\theta_2 = [w_3, b]$ and (ii) four groups of parameters where $\theta_1 = w_1, \theta_2 = w_2, \theta_3 = w_3$, and $\theta_4 = b$. For $\text{S}_d$-SGLD, $\rho = 0.5$ and $K = 4$ was used.

The resulting posterior distributions for $(w_1, w_2)$ and $(w_1, w_3)$ from all five scenarios, with SGLD in the leftmost column as our baseline, can be seen in Fig. 1. We observe that, as expected, correlations between $(w_1, w_2)$ still exist when they are allocated to the same parameter group and become apparently independent when assigned to different groups. We also note that the variance of the distributions shrink as the parameter space is partitioned into smaller groups. The underestimation of posterior variance is a commonly reported finding for VI techniques and is interesting to note that our non-parametric methods appear to exhibit this behavior as well. Finally, it appears that the $\text{S}_d$-SGLD adequately approximates S-SGLD with just slightly higher variances and very minor correlations between parameter groups being exhibited.

## H.2    REAL-WORLD DATA EXPERIMENTS

**Framework details.**    In this subsection, we provide more detailed results for our experiments and a grid search for FMNIST, CIFAR10, and SVHN. We note that all the code apart from the metrics was written in PyTorch (Paszke et al., 2019). Regarding the metrics, ESS was adopted from the TensorFlow probability library (Dillon et al., 2017; Abadi et al., 2016) and IAC was calculated in python. For all the experiments, we used a seed of 2. Moreover, we note that we grouped the parameters in an ordered way for $\text{S}_d$-pSGLD and S-pSGLD. We denoted previously that $K\rho$ is the number of groups. So every parameter will go to the $i \mod K\rho$ group where $i$ is the parameter index. If, for instance, $K\rho$ is 8 then parameter 1 will go to group 1, parameter 2 will go to group 2, parameter 9 will go to group 1, etc. If $K\rho$ is the same as the number of parameters, every parameter will go into its own group.

**MNIST.**    Regarding MNIST, we ran all the experiments for 500 epochs with a batch size of 500 and a learning rate of 1e-2. For $\text{S}_d$-pSGLD, the $K$ is set to 300, which is the forward passes that the model does within 1 epoch. For the grouping of the parameters, for $\text{S}_d$-pSGLD we used group sizes of 2,4,8,32,128,512,2048,4096,8192,16384,32768 and 42200; and for S-pSGLD we used groups sizes of 2,8,32,128,512,2048,4096 and 8192.

**FashionMNIST.**    We ran all experiments for 300 epochs with a batch size of 500. For $\text{S}_d$-SGHMC the $K$ is set to 2, which is the forward passes that the model does within 1 epoch. We observed with experimenting with $K$ that we do not need to set $K$ very high, and even a small number like 2 that we used here is enough to produce the same results as with an $K$ of 200 or 300. In this way, we save significant time in training. Regarding the parameter partitioning, for $\text{S}_d$-SGMCMC, we put every parameter in a different group, and for S-SGMCMC we used groups of 2,4,8, and 16. For $\text{S}_d$-pSGLD, pSGLD, $\text{S}_d$-SGHMC and SGHMC we tested their performances with learning rates of 1e-2,1e-3,1e-4,1e-5. For S-pSGLD we used a learning rate of 1e-3 and for S-SGHMC a learning rate of 1e-2.

Table 4: Evaluation Metrics on FashionMNIST with pSGLD, $S_d$-pSGLD and S-pSGLD

| Method | dropout | LR | IAC | ESS | Accuracy |
|---|---|---|---|---|---|
| $S_d$-pSGLD$_{|\theta|}$ | 0.1 | 1e-05 | 1018 | 5.63 | 0.918 |
| $S_d$-pSGLD$_{|\theta|}$ | 0.1 | 1e-04 | 808 | 7 | **0.925** |
| $S_d$-pSGLD$_{|\theta|}$ | 0.1 | 1e-03 | 754 | 7.48 | 0.924 |
| $S_d$-pSGLD$_{|\theta|}$ | 0.1 | 1e-02 | 723 | 8.05 | 0.911 |
| $S_d$-pSGLD$_{|\theta|}$ | 0.5 | 1e-05 | 778 | 7.08 | 0.923 |
| $S_d$-pSGLD$_{|\theta|}$ | 0.5 | 1e-04 | 777 | 7.15 | 0.923 |
| $S_d$-pSGLD$_{|\theta|}$ | 0.5 | 1e-03 | 737 | 7.57 | 0.924 |
| $S_d$-pSGLD$_{|\theta|}$ | 0.5 | 1e-02 | **677** | **8.24** | 0.91 |
| pSGLD | - | 1e-5 | 779 | 7.09 | 0.924 |
| pSGLD | - | 1e-4 | 774 | 7.16 | 0.911 |
| pSGLD | - | 1e-3 | 770 | 7.26 | 0.809 |
| pSGLD | - | 1e-2 | 745 | 7.48 | 0.724 |
| S-pSGLD$_2$ | - | 1e-3 | 740 | 7.55 | 0.918 |
| S-pSGLD$_4$ | - | 1e-3 | 751 | 7.45 | 0.919 |
| S-pSGLD$_8$ | - | 1e-3 | 776 | 7.24 | 0.919 |
| S-pSGLD$_{16}$ | - | 1e-3 | 855 | 6.64 | 0.916 |

Table 5: Evaluation Metrics on FashionMNIST with SGHMC, $S_d$-SGHMC and S-SGHMC

| Method | dropout | LR | IAC | ESS | Accuracy |
|---|---|---|---|---|---|
| $S_d$-SGHMC$_{|\theta|}$ | 0.1 | 1e-05 | **782** | **7.08** | 0.412 |
| $S_d$-SGHMC$_{|\theta|}$ | 0.1 | 1e-04 | 888 | 6.41 | 0.796 |
| $S_d$-SGHMC$_{|\theta|}$ | 0.1 | 1e-03 | 793 | 6.98 | 0.92 |
| $S_d$-SGHMC$_{|\theta|}$ | 0.1 | 1e-02 | 1113 | 5.06 | 0.922 |
| $S_d$-SGHMC$_{|\theta|}$ | 0.5 | 1e-05 | 790 | 6.93 | 0.207 |
| $S_d$-SGHMC$_{|\theta|}$ | 0.5 | 1e-04 | 789 | 6.9 | 0.758 |
| $S_d$-SGHMC$_{|\theta|}$ | 0.5 | 1e-03 | 796 | 6.81 | 0.0.92 |
| $S_d$-SGHMC$_{|\theta|}$ | 0.5 | 1e-02 | 923 | 5.7 | 0.927 |
| SGHMC | - | 1e-5 | 791 | 6.93 | 0.206 |
| SGHMC | - | 1e-4 | 789 | 6.9 | 0.751 |
| SGHMC | - | 1e-3 | 795 | 6.83 | 0.92 |
| SGHMC | - | 1e-2 | 920 | 5.72 | 0.928 |
| S-SGHMC$_2$ | - | 1e-2 | 928 | 5.67 | **0.928** |
| S-SGHMC$_4$ | - | 1e-2 | 915 | 5.77 | 0.927 |
| S-SGHMC$_8$ | - | 1e-2 | 1142 | 4.87 | 0.919 |
| S-SGHMC$_{16}$ | - | 1e-2 | 1121 | 4.92 | 0.906 |

**CIFAR10.** The setup is similar to the one we used in FashionMNIST as we ran all experiments for 300 epochs with a batch size of 128. For $S_d$-SGHMC, the $K$ is set to 2, which $K$ is the forward passes that the model does within 1 epoch. Regarding the parameter partitioning, for $S_d$-SGMCMC, we put every parameter in a different group, and for S-SGMCMC we used groups of 2,4,8, and 16. For $S_d$-pSGLD, pSGLD, $S_d$-SGHMC and SGHMC we tested their performances with learning rates of 1e-2,1e-3,1e-4,1e-5. For S-pSGLD, we used a learning rate of 1e-3, and for S-SGHMC, a learning rate of 1e-2. We focused our strategy on evaluating the accuracy of the different combinations of hyperparameters with the proposed methods, as can be seen in Figs. 4 and 5. Quantitative results on IAC, ESS and maximum accuracy are depicted in Tables 6 and 7.

Table 6: Evaluation Metrics on CIFAR10 with pSGLD, $S_d$-pSGLD and S-pSGLD

| Method | dropout | LR | IAC | ESS | Accuracy |
|---|---|---|---|---|---|
| $S_d$-pSGLD$_{|\theta|}$ | 0.1 | 1e-02 | 623 | 7.23 | 0.191 |
| $S_d$-pSGLD$_{|\theta|}$ | 0.1 | 1e-03 | 572 | 7.6 | 0.896 |
| $S_d$-pSGLD$_{|\theta|}$ | 0.1 | 1e-04 | 692 | 6.45 | 0.921 |
| $S_d$-pSGLD$_{|\theta|}$ | 0.1 | 1e-05 | 922 | 4.88 | 0.922 |
| $S_d$-pSGLD$_{|\theta|}$ | 0.5 | 1e-02 | **546** | **8.01** | 0.768 |
| $S_d$-pSGLD$_{|\theta|}$ | 0.5 | 1e-03 | 582 | 7.88 | 0.918 |
| $S_d$-pSGLD$_{|\theta|}$ | 0.5 | 1e-04 | 691 | 6.85 | 0.926 |
| $S_d$-pSGLD$_{|\theta|}$ | 0.5 | 1e-05 | 620 | 7.22 | **0.927** |
| pSGLD | - | 1e-2 | 716 | **8.01** | 0.666 |
| pSGLD | - | 1e-3 | 740 | 7.87 | 0.866 |
| pSGLD | - | 1e-4 | 780 | 7.41 | 0.914 |
| pSGLD | - | 1e-5 | 831 | 6.89 | 0.926 |
| S-pSGLD$_2$ | - | 1e-3 | 600 | 7.44 | 0.894 |
| S-pSGLD$_4$ | - | 1e-3 | 599 | 7.4 | 0.905 |
| S-pSGLD$_8$ | - | 1e-3 | 709 | 6.41 | 0.881 |
| S-pSGLD$_{16}$ | - | 1e-3 | 767 | 5.93 | 0.836 |

Table 7: Evaluation Metrics on CIFAR10 with SGHMC, $S_d$-SGHMC and S-SGHMC

| Method | dropout | LR | IAC | ESS | Accuracy |
|---|---|---|---|---|---|
| $S_d$-pSGLD$_{|\theta|}$ | 0.1 | 1e-02 | 608 | 7.16 | 0.91 |
| $S_d$-pSGLD$_{|\theta|}$ | 0.1 | 1e-03 | 975 | 4.6 | 0.922 |
| $S_d$-pSGLD$_{|\theta|}$ | 0.1 | 1e-04 | 654 | 6.63 | 0.869 |
| $S_d$-pSGLD$_{|\theta|}$ | 0.1 | 1e-05 | 652 | 6.65 | 0.724 |
| $S_d$-pSGLD$_{|\theta|}$ | 0.5 | 1e-02 | 584 | 7.7 | 0.918 |
| $S_d$-pSGLD$_{|\theta|}$ | 0.5 | 1e-03 | 751 | 6.23 | **0.927** |
| $S_d$-pSGLD$_{|\theta|}$ | 0.5 | 1e-04 | 679 | 6.73 | 0.886 |
| $S_d$-pSGLD$_{|\theta|}$ | 0.5 | 1e-05 | 772 | 6.01 | 0.778 |
| pSGLD | - | 1e-2 | 727 | **7.94** | 0.86 |
| pSGLD | - | 1e-3 | 832 | 6.84 | 0.926 |
| pSGLD | - | 1e-4 | 862 | 6.57 | 0.885 |
| pSGLD | - | 1e-5 | 858 | 6.6 | 0.746 |
| S-pSGLD$_2$ | - | 1e-3 | **583** | 7.49 | 0.913 |
| S-pSGLD$_4$ | - | 1e-3 | 624 | 7.03 | 0.919 |
| S-pSGLD$_8$ | - | 1e-3 | 904 | 4.97 | 0.908 |
| S-pSGLD$_{16}$ | - | 1e-3 | 822 | 5.47 | 0.774 |

**SVHN.** We also ran all of the experiments for 300 epochs with a batch size of 128. Here for $S_d$-SGHMC, the $K$ is set to 2, which is the forward passes that the model does within 1 epoch. We note that $K$ here is less than on CIFAR10 and FashionMNIST, but as we mentioned before, this does not make a difference for our results, as we have tested. Regarding the parameter partitioning, for $S_d$-SGMCMC, we put every parameter in a different group, and for S-SGMCMC we used groups of 2,4,8, and 16. For $S_d$-pSGLD, pSGLD, $S_d$-SGHMC and SGHMC we tested their performances with

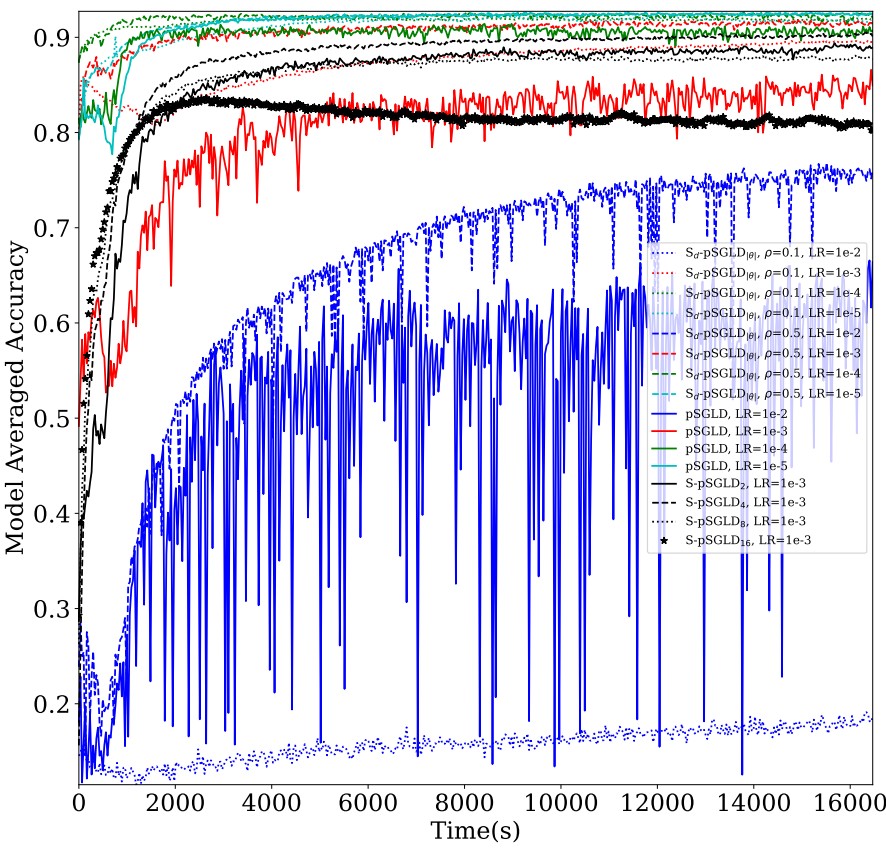

Figure 4: Model comparison with different hyperparameters for the pSGLD versions of our baselines and proposed algorithms. We evaluated expected accuracy over iterations on CIFAR-10. Subscript "d" denotes the dropout version of our approach. LR refers to the learning rate, and $\rho$ denotes the dropout rate.

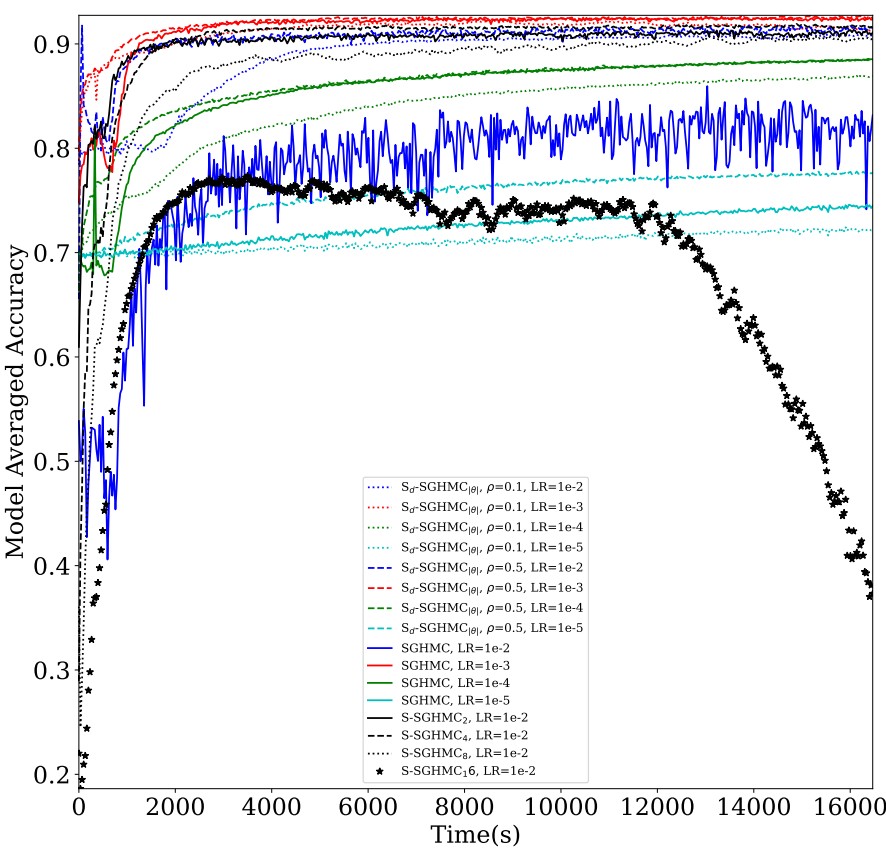

Figure 5: Model comparison with different hyperparameters for the SGHMC versions of our baselines and proposed algorithms. We evaluated expected accuracy over iterations on CIFAR-10. Subscript "d" denotes the dropout version of our approach. LR refers to the learning rate, and $\rho$ denotes the dropout rate.

learning rates of 1e-1,1e-2,1e-3,1e-4,1e-5,1e-6. For S-pSGLD we used a learning rate of 1e-4, and for S-SGHMC, a learning rate of 1e-2. Same as in CIFAR10, we conducted a grid search for learning rate, dropout rate, and optimizers to find the best performing models and test them for their accuracy. We can observe these results in Figure 3 in the main paper. The strategy that we followed is the same as in CIFAR10 and is presented in Figs. 6 and 7.

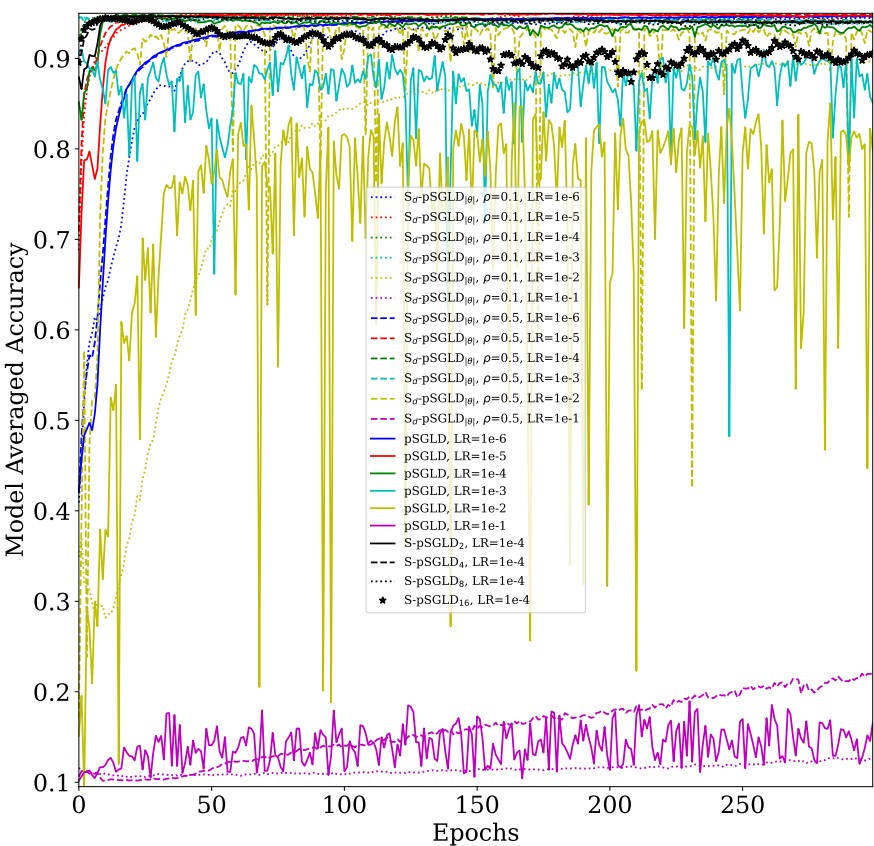

Figure 6: Model comparison with different hyperparameters for the pSGLD versions of our baselines and proposed algorithms. We evaluated expected accuracy over iterations on SVHN. Subscript "d" denotes the dropout version of our approach. LR refers to the learning rate, and $\rho$ denotes the dropout rate.

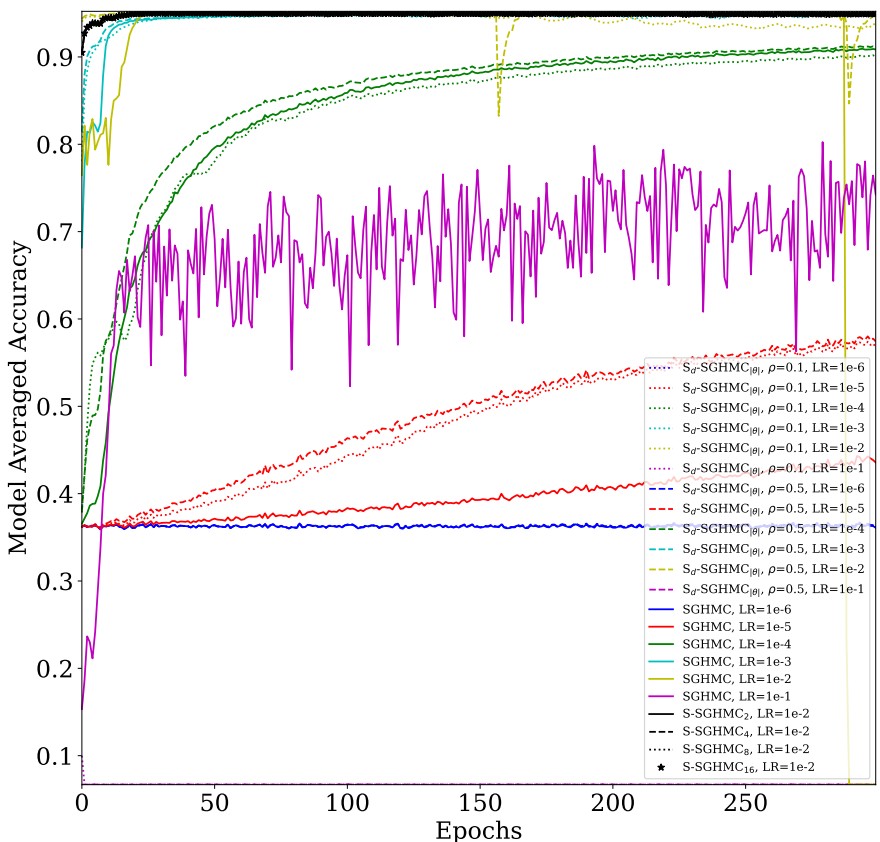

Figure 7: Model comparison with different hyperparameters for the SGHMC versions of our baselines and proposed algorithms. We evaluated expected accuracy over iterations on SVHN. Subscript "d" denotes the dropout version of our approach. LR refers to the learning rate, and $\rho$ denotes the dropout rate.

