# OpenReview forum: "Structured Stochastic Gradient MCMC"
_ICLR.cc/2022/Conference — ICLR 2022 Submitted_

### Official Review · Reviewer_AW5Q · 2021-10-22

**Correctness:** 3
**Technical Novelty And Significance:** 3
**Empirical Novelty And Significance:** 2
**Recommendation:** 5
**Confidence:** 4

**Main Review:**

## Strength
Personally, I find the proposed method interesting. The paper is written clearly and easy to follow, and the overall idea is easy to understand. I have checked the proofs of the theorem and they seem to be correct. To support the claims, the proposed methods were applied to real-world large data set to confirm their advantages compared to the standard full SGMCMC with some ablation studies, although I still think there is room for improvement.

## Weakness and concerns
Although the proposed method is easy to follow, I still find some parts unclear and elaborations are needed. First, the author claims that building the independent structure into the posterior can help improve the mixing speed (bottom of page 1, the first paragraph in the conclusion, etc.). However, I cannot see direct reasoning behind those claims, could you elaborate more on this? In section 6.1, I am not sure I understand the reasoning behind the better performances of S_d-SGMCMC. Can you elaborate more on "regularizing the model"?
On page 4, why $\tilde{\theta}^{(t,i)}$ is composed of samples from previous timesteps? I thought the author mentioned that a **single** sample $\tilde{\theta}^{(t,i)}$ from the **current** timestep is used for Monte Carlo approximation. I recommend putting more details in algorithm 1. For example, elaborating how $\tilde{\theta}^t_{\neg i}$ is drawn from $\hat{q}^t_{\neg i}$.
In addition, why do appendix B and D use the same title? Can you merge them together?

Another concern is the theoretical soundness of proposed algorithms. For S-SGMCMC, the Monte Carlo estimation of the modified energy function requires previous samples. Does this break the Markov assumption of SGMCMC since it should only depend on the **current** samples? For example, when the sampler does not reach the stationary stage, $\hat{q}^t$ still evolves with time, thus, the samples from the previous timesteps are not from $\hat{q}^t$.
Apart from the stationary distribution of S-SGMCMC, I also wonder what is the stationary distribution (if exists) of the dropout version? How different is the stationary distribution of the dropout version compared to the optimal $q$? Any theoretical guarantees on the correctness of the dropout version?

In terms of the empirical evaluation, I wonder about the uncertainty quantification ability of the proposed algorithms, which is an important metric for SGMCMC methods, especially since the proposed methods also seem to underestimate the posterior variance. In addition, in the abstract, the author mentioned better predictive likelihoods. However, I can only find the accuracy metric in the experiment, which is different from the predictive likelihood.





**Summary Of The Paper:**

The author proposed a framework to incorporate the independence structure into the posterior inference for faster-mixing speed. To achieve that, the author designed two specific algorithms called S-SGMCMC and S_d-SGMCMC, respectively. Specifically, S-SGMCMC consisted of the following steps. First, the target random variables ($\theta$) are gathered into mutually independent groups. Then, a modified energy function is derived by minimizing the KL divergence between the posterior $q$ and target $p(\theta|D)$. The last step is to apply a standard SGMCMC method to draw samples from the resulting modified energy function. Further, the author also built a connection between dropout and the modified energy function, which results in a structure dropout SGMCMC (S_d-SGMCMC) with better scalability.
The author claimed the resulting algorithms achieved faster mixing speed and better classification accuracy when applied to real-world classification tasks.


**Summary Of The Review:**

The paper presented an interesting approach to incorporate independent structures into posterior inference. However, there are still some ambiguities in motivation and theoretical soundness. To better support the claims, some metrics or experiments regarding uncertainty quantification should be added or discussed since the proposed methods also seem to underestimate the variance.

---
I have read the author's responses. It address some of my concerns. However, for my last concern regarding the uncertainty (which is one of the reasons we consider MCMC over variational inference), the author did not mention it at all. Since this algorithm also underestimates the uncertainty due to minimizing the exclusive KL, I recommend the author to demonstrate that it can provide reasonable uncertainty quantification. Therefore, I will keep my original evaluations.

---

### Official Review · Reviewer_qXUG · 2021-11-02

**Correctness:** 4
**Technical Novelty And Significance:** 3
**Empirical Novelty And Significance:** 1
**Recommendation:** 8
**Confidence:** 5

**Main Review:**

This is a very well-written, clear paper, with nice experiments to guide intuition. Congratulations! The effort shows.

If the paper "attempt[s] to hybridize MCMC and VI", why not compare to structured variational inference? Is the only difference between this approach and variational inference the additional noise in the gradient update?

If so, it could be stated in a sentence, which would help me be less confused. After all, Equation (4) is the same objective function, theorem 1 is roughly the same proof as coordinate ascent variational inference (where the parameter groups M are simply coordinates i), then these are more of an 'observation' than a theorem. But the key insight of using the coordinate ascent variational inference for SGMCMC is valuable and the main point of thh paper, and calling it a theorem may be helpful for readers. So it's up to the authors for deciding what's best - just wanted to point out that these connections and derivation analogs could be made more clear, which would make the method easier to understand and exposition more straightforward.

- The most confusing sentence for me was belot Eq. 6: "in practice, at timestep t...". Maybe describe how \hat q is composed of samples from previous timesteps.

- small wording choices: "completely joint", "well-approximates", "solid distributional assumptions"

- in equation 8, rho_i is not defined

I also appreciate the authors including clean code as a supplementary zip file for reproducibility.


**Summary Of The Paper:**

This work proposes using a structured variational approximation for stochastic gradient Markov Chain Monte Carlo. This allows someone to choose a factorization for the variational distribution (which factorization is best is unclear; several are studied). Analogously to coordinate ascent variational inference, the authors show that the best approximation is the Boltzmann energy function marginalized over the complements of every parameter group.

However, this structured approximation is computationally expensive and requires the same number of evaluations of the approximation as there are parameter groups. This computational burden is alleviated by a dropout scheme, where instead of sampling from every parameter groups, parameters are masked using a dropout distribution, and the number of stochastic masks is a hyperparameter that controls regularization and fidelity to the structure imposed in the factorization.

Experiments show that this is a viable way to impose structure on a variational distribution, and that mixing times are improved.

**Summary Of The Review:**

This is a clearly written paper with a tightly scoped contribution. I advocate for acceptance, and looking forward to follow up work.

---

### Official Review · Reviewer_zQoE · 2021-11-02

**Correctness:** 1
**Technical Novelty And Significance:** 2
**Empirical Novelty And Significance:** 2
**Recommendation:** 3
**Confidence:** 4

**Main Review:**

1. The proposed algorithm is not well motivated: the VI approach aims to obtain a point estimate, SGMCMC aims to draw a sequence of samples from the target posterior, but the proposed algorithm is to simulate a sequence of samples from a modified distribution.
It is unclear how well the modified distribution approximates the target distribution and how useful the pseudo-posterior samples are for statistical inference of the target distribution.

2. The comparison with the existing SGMCMC algorithm is not convincing. It seems that they are ``comparable in wall clock time'', while the baselines pSGLD and SGHMC might not be the state-of-the-art.

**Summary Of The Paper:**

This paper proposes a hybrid SGMCMC algorithm, a Langevin-type algorithm that operates on a modified energy function based on the variational inference (VI) approach.

**Summary Of The Review:**

This paper proposes a hybrid SGMCMC algorithm, but the underlying theory is not fully developed and the performance of the algorithm is not fully explored.

---

### Official Review · Reviewer_NCgf · 2021-11-02

**Correctness:** 2
**Technical Novelty And Significance:** 3
**Empirical Novelty And Significance:** 2
**Recommendation:** 5
**Confidence:** 3

**Main Review:**

Overall I enjoyed reading the paper and I think the proposed method has merit, but I also do have some concerns.

1)	The paper does not have proof of convergence for the proposed algorithm, a proof of convergence would have made the paper stronger.

2)	Although the paper deals with multivariate models and a potentially large number of parameters, the evaluation metrics are univariate. For example, the ESS in table 1. The authors should report the results for multivariate ESS, especially considering that on, for example, CIFAR-10 the results are only marginally better than SGHMC.

3)	Please note, that in the recent paper [1] it was shown that ESS and “other standard MCMC metrics which do not account for sample bias are not appropriate diagnostic tools for SGMCMC”. This paper [1] proposes to use the kernel Stein discrepancy metric for the assessment of SGMCMC methods. I would recommend the authors include it in their analysis.

4)	On page 4 the authors assume that “This partitioning structure is assumed to be known a priori.” when talking about factorization of the parameters into mutually independent groups. While it can be naturally assumed in some models, is this a straightforward assumption of BNN, since the proposed algorithm is in particular motivated by application to BNN?

5)	On page 4 the authors mention: “While it is unlikely to have the procedure initialize to a stationary state, we observe in practice that our scheme both tends to converge towards and remain in a stationary state. “ I would encourage the authors to directly refer to a figure or table which illustrates this claim.

6)	On page 3 it states: “The gold standard for approximating the entire posterior distribution is by deploying Markov chain Monte Carlo (MCMC) algorithms. These methods work by producing an empirical distribution of samples through a random walk in parameter space. “
I would argue that this is not a correct statement as not all MCMC methods deploy random walk behavior (for example, not HMC or NUTS).

7)	In the appendix on page 17 you mention “We observe that as we break dependencies we capture similar uncertainty intervals. ”. To me it looks like the confidence intervals actually change quite a bit, especially left/right of figure 4a and left/right of figures 4b and 4c, the confidence intervals become quite a bit more narrow.  Can this again be attributed to VI behavior?

8)	In Figure 6 the average accuracy for the last model in the list is quite surprising. Do you have any intuition about that?

9)	Please, check your references, both formatting, and whether some of the papers you cite have been published in the meantime.

10) On page 3 the sentence " The answer is affirmative and will be answered as follows" probably can be re-written in a better way.

[1] Nemeth, Christopher, and Paul Fearnhead. "Stochastic gradient Markov chain Monte Carlo." Journal of the American Statistical Association 116.533 (2021): 433-450.

UPD: I increased my score for correctness to 3 after the authors reply and overall score to 5.

**Summary Of The Paper:**

The paper proposes a new hybrid method between MCMC and VI. The main idea of the paper is construction of a new energy function which allows to speed up the sampling significantly in comparison to the SGMCMC (stochastic gradient MCMC). An additional modification of the proposed algorithm is done by adopting a drop-out inspired approximation which allows for even better scalability.

**Summary Of The Review:**

The paper takes a relevant direction in research trying to make inference more scalable for BNN (and other models) by combining MCMC and VI in a stochastic gradient MCMC framework. The paper, however, does not provide theoretical guarantees of convergence while empirical evaluation metrics were not chosen optimally (only univariate metrics which are also not optimal for SGMCMC methods as shownin [1]). Consequently, I would encourage the authors to address the critical points and resubmit the paper in the future.

[1] Nemeth, Christopher, and Paul Fearnhead. "Stochastic gradient Markov chain Monte Carlo." Journal of the American Statistical Association 116.533 (2021): 433-450.

---

### Decision · Program_Chairs · 2022-01-20

**Decision:**

Reject

**Comment:**

The authors propose a flexible variational posterior approximation, relaxing unrealistic factorization and strong parametric constraints that are standard. There was a mixed reception from reviewers. Overall, the paper is on the borderline. The presentation and empirical investigation could be changed so that the nice contributions in the paper are more easily recognized. Indeed, after rebuttal several reviewers still felt like their concerns were not fully addressed. One reviewer was concerned about the evaluation metrics, and wanted to see Stein discrepancy instead of ESS, and did not feel the ESS was sufficiently motivated (as described in updated comments). Another reviewer felt the uncertainty of the predictive distribution was sufficiently well evaluated. Another reviewer generally satisfied by the response. The decision could go either way, but the paper would probably be more widely appreciated by a significant revision, carefully taking into account the questions of the reviewers. The authors are encouraged to accommodate reviewer questions in future versions of the paper.